# Understanding Urban Land Leasing System as a Strategic Value Capture Instrument to Enhance Urban Revenue in Ethiopia: A Case Study of Bahir Dar City

Seid Hussen Yimam [1,*] , Hans Lind [2] and Belachew Yirsaw Alemu [1]

1   Institute of Land Administration, Bahir Dar University, Bahir Dar 5001, Ethiopia; belachew.y02@gmail.com
2   Department of Real Estate and Construction Management, KTH—Royal Institute of Technology,
    100 44 Stockholm, Sweden; hanslin@kth.se
*   Correspondence: yimamse@gmail.com; Tel.: +251-918768610

**Abstract:** Since 1993, the Ethiopian government has been using the urban land leasing system to monetize the increase in land value created due to factors other than private investment. Thus, this paper aims to explore and understand whether Bahir Dar city is leveraging the urban land lease system as a strategic value capture instrument to enhance its local revenue or not. This study has used the qualitative research method and in-depth analysis. It has used a desk review of documents and key informant interviews of experts and brokers to collect the data required to realize the objective of this study. The study has found that the majority of the urban land is held in a permit system, with landholders paying a small amount of land rent per annum. In addition, the study has uncovered that the city has transferred most of its land through administrative allotment at low and outdated benchmark prices, which has dwindled its lease revenue. Moreover, it has identified that there is weak enforcement of the lease payment collection, and it has adversely affected the city's urban revenue generation potential. As a result, the city is not leveraging the public land leasing system as a strategic value capture instrument. Based on these findings, this paper has advised the government to implement a modern property tax system to capture value increment on permit holding land. In addition, the study proposes to carry out an empirical investigation and identify the factors that significantly affect the benchmark prices and update the benchmark price regularly given those factors. Moreover, the study has suggested a proper enforcement of the lease payment collection in the city.

**Keywords:** urban land lease; value capture; lease benchmark price; urban revenue; land lease tender price; land lease payment

## 1. Introduction

Different writers have argued that urbanization drives economic growth in any country (Montgomery et al. 2013; Bahl and Linn 2014; Palmer and Stephen 2015; Zhang and Xu 2016; Zhang et al. 2020), since most of the economic actors that encourage economic growth are mainly located in cities (Bahl and Linn 2014). In sub-Saharan Africa (SSA), however, cities have been growing without associated economic growth. Cities are not the gist of economic growth in this region. Instead, they are becoming the causes of economic and social problems in the region (Montgomery et al. 2013). The rapid rate of urbanization in the SSA has increased the demand for urban infrastructures and other public services. The availability of urban infrastructure facilities is a sine qua non to the development of urban areas. However, several cities in SSA, are characterized by fast urbanization and declining infrastructural services (Sietchiping et al. 2012). Collier and Venables (2016) also pointed out that about 80% of the people, in the cities of the region, are living in informal houses where there are no proper urban infrastructure facilities. There is an infrastructural gap in the cities of SSA due to a shortage of local revenue (Fjeldstad and Heggstad 2012). Oualalou

(2012) also pointed out that the provision of urban infrastructure and other crucial public services requires an estimated investment of more than 90 billion US dollars per year in the SSA.

However, the region has so far mobilized only half of this required budget from its sources (Oualalou 2012). This is because the traditional own sources of revenue are not enough to generate the required finance. As a result, many cities in SSA experience fiscal deficits (Berrisford et al. 2018).This situation demands the cities in the SSA region to think of innovative revenue sources. Studies (Blanco Blanco et al. 2016; Berrisford et al. 2018; Walters 2012) indicate that different cities across the world use land value capture (LVC) as an innovative instrument to collect land value created by municipal investments. The theory of land value capture asserts that the revenue gathered from land value appreciation created by public investment can cover the cost of infrastructure and public services (Kresse et al. 2020). Worldwide, theoretical and empirical literature uncovered that different countries use public land leasing as a value capture instrument to enhance local revenue. For instance, the experiences of China and Hong Kong show that they have been utilizing the system of land-based financings, such as public land leasing, to cover the cost of urban infrastructure (Peterson 2007). Studies carried out in China (Ye and Wu 2014; Zhang and Xu 2016; Zhou et al. 2017) asserted that land lease revenue has been playing a great role in providing finance to facilitate urban development. Mainly collected in a lump sum, land lease revenue has become the main source of local finance in Hong Kong (Perkins 2009). In addition, Yang et al. (2015) stated that the increase in competition, which was created due to fiscal decentralization, motivated local governments to optimally utilize land leasing to increase their local revenue. Similarly, studies reveal that different African nations (such as Ethiopia, Botswana, Ghana, Zambia, Rwanda, etc.) have been applying urban land leasing system as a means of generating land-based revenue, along with its other development objectives (Peterson 2007; Kopanyi 2014; Goodfellow 2017; Adams et al. 2003; UN-Habitat 2014).

Ethiopia, one of the SSA countries, is located in the Horn of Africa, with a total area of about 1.128 million square kilometers. According to the UN (2019) World Urbanization Prospects, the population of Ethiopia is forecasted to grow from 108 million in 2018 to 191 million in 2050. The proportion of the urban population is also forecasted to increase from 21% (approximately 23 million) in 2018 to 39% (75 million) in 2050. The country has been implementing urban land rent and house tax as land value capture instruments to gather urban revenue during the Derg or Socialist regime as per proclamation No. 80/1976. After it came into power in 1991, the Transitional Government of Ethiopia (TGE) found that the urban revenue collected from land rent, house taxes, and other sources was not adequate to finance the provision of infrastructure and other public services. Besides, it realized that the existing law could not enable municipalities to monetize the increase in land value created due to public investment and economic growth. Consequently, the TGE introduced the urban land leasing policy in 1993, as per proclamation no. 80/1993, which aimed to enable urban centers to improve their local revenue and monetize the land value appreciation unjustifiably collected by private landholders, among its other objectives (FDRE 1993). With this reform, the TGE separated land ownership rights from land use rights in the urban areas.

Afterwards, the government has undertaken a sequence of amendments to the urban land lease proclamation. The first proclamation allowed public tendering as the only mode of urban land leasing (FDRE 1993). Since the proclamation did not apply to the urban land held during the Derg regime (FDRE 1993), two different forms of urban land tenure (permit and lease) co-existed in the country, which led to "tensions between *de jure* and *de facto* rights of urban land" (Adamu 2014). In addition, studies found that the tender method resulted in a rigid urban land supply (Yusuf et al. 2009). Thus, in 2002, the government enacted proclamation no. 272/2002 to fill the existed gaps, and this replaced proclamation no.80/1993 (Adamu 2014; Alemu 2010). The new proclamation stipulated how to convert the urban land held through a permit system into a lease system (FDRE 2002). Besides, the decree introduced negotiation as an additional method of land leasing.

However, the negotiation method opened opportunities for corrupted practices (Yusuf et al. 2009). Studies (Adamu 2014; Yusuf et al. 2009) show municipalities experienced ineffective, corrupted, and less transparent land delivery practices. Due to this and other compelling reasons, the government replaced proclamation no.272/2002 with proclamation no.721/2011, considering good governance as the fundamental prerequisite to realize an efficient, transparent, and well-functioning land market (Adamu 2014).

Many studies have been conducted on urban land lease policy issues in Ethiopia. Some studies have assessed the performance of the urban land management system and the effect of the urban land lease system in providing land for investment and business expansion (Yusuf et al. 2009; Gebremariam and Mailimo 2016; Alemu 2010; Belete 2017), and quantitatively analyzed the difference between benchmark price and land auction winning price in Addis Ababa (Weldesilassie and Gebrehiwot 2017). The other studies have analyzed the shortcomings and effects of the urban land lease policy reforms of Ethiopia (Adamu 2014), the current urban land lease holding proclamation from the perspective of tenure security and property rights (Tigabu 2014), and the impact of the current urban land lease policy on the freedom and equity of transferring land user rights (Ambaye 2016). In a study comparing Kigali and Addis Ababa cities, Goodfellow (2017) has studied the politics of urban land value capture. He asserted that Addis Ababa has become relatively successful in raising land-related revenue using public land leasing as the main value capture instrument. Questioning the sustainability of the public land leasing in generating revenue, the writer suggested the government to introduce a property tax system as an alternative source of urban revenue (Goodfellow 2017). In addition, Gebrihet and Pillay (2020) have quantitatively investigated the determinants of the urban land lease market in Ethiopia, focusing on Mekelle city.

Teferi (2009) also assessed the implementation of the urban land lease policy in Bair Dar city and identified intervention, lack of transparent decision-making, absence of auditing, and lack of taking corrective measures as the main factors that adversely affect lease revenue collection in the city. In a study carried out in some cities of Amhara and Tigray regions, Peterson (2007) found that public urban land leasing contributes between 21% and 45% of the total municipal revenue from 2003 to 2004, with Bahir Dar city registered the highest proportion. The study pointed out that the country could further increase its urban revenue, from urban land leasing, by improving the entrepreneurial mindset of the city administration and encouraging upfront lease payment. However, this study has ignored the other crucial factors that may affect the public land leasing system as a value capture instrument. In a municipal revenue baseline survey undertaken at Dire Dawa, Mekelle, and Bahir Dar city, MoUDH (2016) found that urban land leasing revenue, on average, contributes 27%, 45%, and 56% of the total municipal revenue in Dire Dawa, Mekelle, and Bahir Dar from 2010/11 to 2014/15, respectively. Both studies have found that Bahir Dar city is better at utilizing urban land leasing to enhance its urban revenue compared with the other cities studied. However, MoUDH (2016) has uncovered that Bahir Dar city faces a fiscal deficit in financing infrastructure and other public services delivery given its huge land-based revenue generation potential.

The reviews of the literature show that most of the studies (Yusuf et al. 2009; Alemu 2010; Gebremariam and Mailimo 2016; Belete 2017) have paid attention to assessing the impact of the urban land lease policy on real estate investment, private sector business expansion and property rights in Addis Ababa and the others (Adamu 2014; Tigabu 2014; Ambaye 2016) are national-level studies. However, Goodfellow (2017) has tried to analyze public land leasing from the perspective of value capture although he has focused on Addis Ababa, and this might not reflect the situation in Bahir Dar city due to differences in location and socioeconomic conditions between the two cities. In addition, Teferi (2009) has not exhaustively addressed the factors that may affect the urban land leasing system in generating local revenue. To sum up, the previous studies have paid little attention to urban land leasing from the context of value capture in Ethiopia. As a result, there are no adequate documented scientific studies carried out on the extent to which a city such as Bahir Dar

city has utilized urban land lease as a strategic value capture instrument to enhance its local revenue although it is an essential public policy issue. These circumstances, therefore, can justify undertaking an in-depth investigation and detailed analysis to understand the implementation of the urban land leasing system from the perspective of value capture and revenue enhancement in Bahir Dar city.

Thus, the purpose of this study is to explore and understand whether Bahir Dar city is leveraging the urban land lease system as a strategic value capture instrument to enhance its local revenue or not. The main contribution of this study is to provide decision-makers and practitioners with an in-depth understanding and some scientific insights for improving the existing urban land leasing system as a strategic value capture instrument. Based on this objective, this research has addressed the following questions: (1) How can the urban land leasing system enable the government to capture the upsurge in land value realized from public investment, economic growth, and population growth in Bahir Dar city? (2) Is the city leveraging the urban land leasing system to enhance its local revenue? (3) What are the main factors that deter the city from enhancing its revenue related to public land leases? (4) What are the actions that should be taken to enable the system to improve the city's local revenue?

The remaining part of the paper is organized as follows. Section 2 discusses the research methods and the study site. Section 3 covers the theoretical framework. In Section 4, we present the findings and discussions from the case study. Finally, Section 5 addresses the conclusion and policy implications of the study.

## 2. Methods and Study Site

### 2.1. Methods

The case study design is the most relevant research design when the study addresses the how, why and what research questions; when the investigator has emphasized understanding the contemporary situations existing within a particular real-life context, and when the issue needs extensive description and in-depth understanding (Yin 2003, 2018). Researchers can explore real-time phenomena using a case study method, which is the most commonly used method in qualitative research (Rashid et al. 2019). Thus, a case study design is selected as more relevant for undertaking this study due to the following reasons. First, it is impossible to cover all cities of the country in this study since it is difficult for the researcher to control the events in such different places. Second, this study has included the 'what' questions related to the factors that hinder the city from optimally utilizing the urban land leasing system as a strategic value capture instrument. Third, location-specific factors affect land lease revenue, and it needs a case study investigation. Bahir Dar city was selected as a case study area because of the following reasons. Firstly, the legal, social, and economic characteristics of the city enables it to be representative of the emerging Regio-Politan cities in the country. Secondly, it is one of the larger regional cities, and it has been implementing the urban land leasing system in the country since 1993. Third, the Federal government of Ethiopia has selected Bahir Dar city as one of three pilot cities to test the implementation of the urban revenue reform project in the country.

This research has used a qualitative research approach whereby the preliminary observation has been followed by a detailed exploration of the urban land leasing implementation, from the perspective of value capturing, in Bahir Dar city. Both secondary and primary data were used to realize the objectives of the study. In a case study research, researchers can use different data collection instruments such as semi-structured interviews together with observation and document collection (Rashid et al. 2019). Gathering data from several sources can help with triangulation and improve the depth of the study (Rashid et al. 2019; Yin 2018).

The researchers collected the secondary data from published and unpublished materials. Public policy statements, development programs, and legal documents were reviewed to understand their implication for the implementation of the urban land leasing system in the city. The 7 consecutive years (from 2013/14–2019/20) of land lease revenue, total

municipal revenue, and land lease tender price of the city were gathered from the annual reports of the revenue office, the office of finance and economic cooperation, and housing construction & development office of the city. In addition, books, journals, and newspapers prepared by Universities, research institutions, and national, regional, and international organizations were reviewed.

The primary data were collected from first-hand sources, key informants, using the semi-structured interview guides. Two forms of interview guides were prepared: one for experts and officials of government offices, and the other for brokers in the real estate market. The interview guides addressed issues such as the process of land leasing, estimation of benchmark prices, collection of lease payments, land use rights transactions, etc. A total of 18 interviewees were purposively selected: 15 from Amhara National Regional State (ANRS) bureau of urban development and construction, ANRS Bureau of revenue, the urban revenue reform project Bahir Dar branch office, and different relevant departments of Bahir Dar city municipality, and 3 from real estate brokers in the city. The interviewees were selected considering their experience. The face-to-face key informants' interviews were administered by the authors and took between two and three hours each.

The data were qualitatively analyzed in such a way that they were reduced and displayed, and then conclusions were drawn from them. Initially, preliminary codes were given, based on the information obtained from the related literature, research objectives, and research questions. Then, the data were reduced, unpacked, and classified into different categories or themes according to the assigned codes. Categories and subcategories were developed based on issues such as the mode of land allocation, determination of benchmark lease price, method of collecting lease payments, trends of change in land lease tender price, the contribution of lease revenue to the total municipal revenue of the city, and enforcement of lease payment collection in the city. To summarize, the qualitative data were analyzed in the form of words or texts, in an iterative way starting from the stage of data collection up to the conclusion.

### 2.2. Study Site

Bahir Dar city, which is the capital of Amhara National Regional State (ANRS), is categorized as Regio-Politan city administration as per ANRS Proclamation No.245/2017 (ANRS 2017). It has urban and peri-urban (rural) *kebeles*. With an area of 16,000 hectares, considering the city proper (MoUDH 2019), the city comprises 6 sub-cities and 25 *kebeles*. *Kebele* represents the smallest administrative unit in the country. The city embraces the nearby small towns of *Tis Abay*, *Meshneti*, and *Zenzelema*, as Satellite towns based on the information obtained from Bahir Dar city administration in 2020. According to Ethiopia's Central Statistical Agency (CSA 2013) projection, the city has an estimated total population of 362,297, of whom 87% live in the urban area, and the remaining 13% reside in the peri-urban (rural) *kebeles* in 2015. Since Bahir Dar city has been authorized to collect and utilize revenue from public urban land leasing and other revenue streams, as per ANRS Regulation No. 135/2015, it has been using urban land leasing as one of the main sources of local revenue (ANRS 2015). Under its role as the capital city of ANRS and its proximity to Lake Tana, Bahir Dar city has become the main hub of government offices, different services centers, and the tourist center of the region. Similar to the cities of other developing nations, the population of Bahir Dar city has been rapidly increasing, mainly due to rural-urban migration. The map of the city is shown in the following figure (Figure 1).

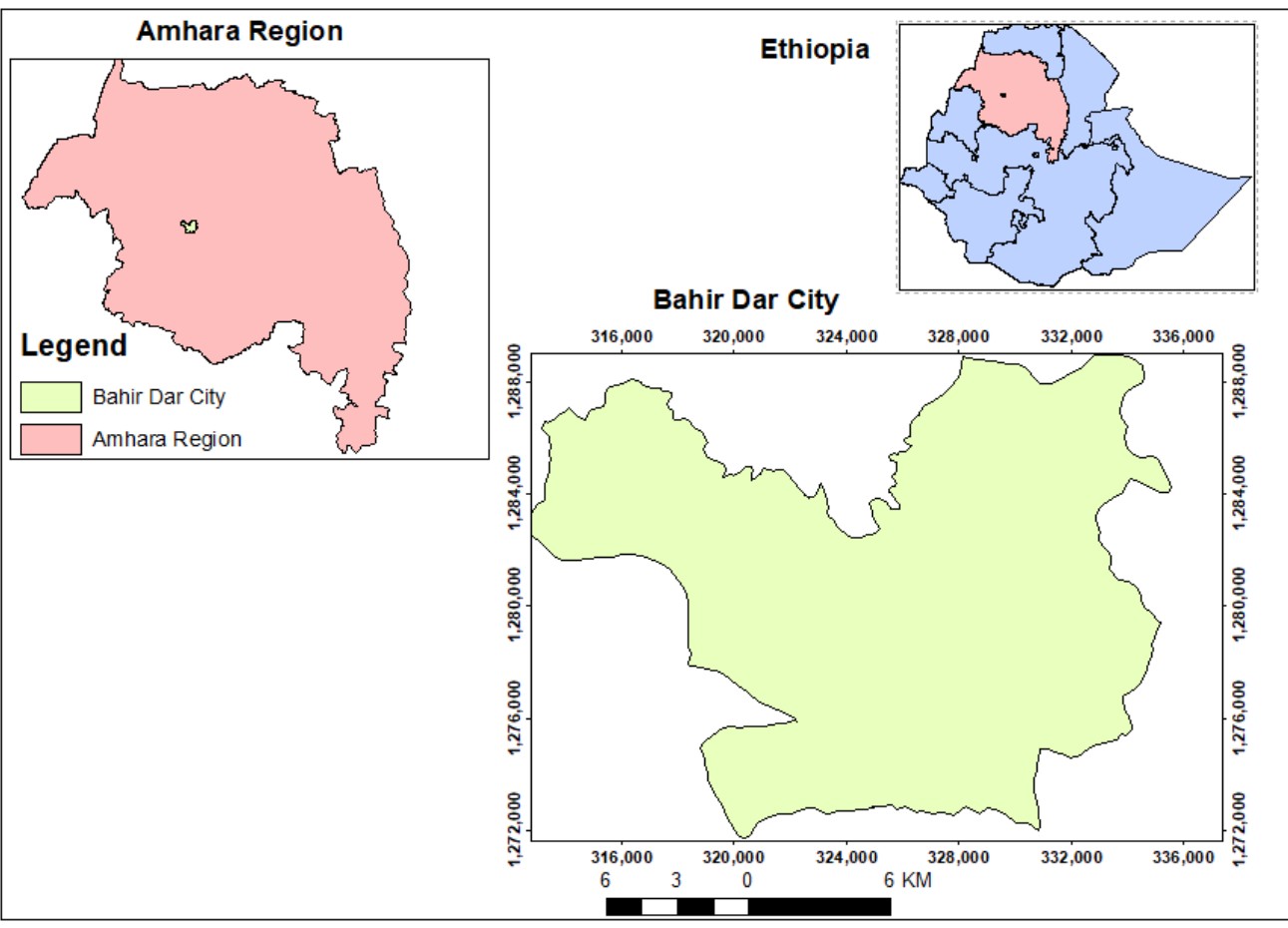

**Figure 1.** Map of Bahir Dar City. Source: Constructed by the author.

### 3. Theoretical Framework

*3.1. The Concept of Land Value Capture*

Suzuki et al. (2015) stated that the concept of land value capture originated from the idea of David Ricardo and Henry George. George understood that as the demand for land increased, due to population growth, landowners appropriated larger income from their landholding, without making any investment in it. Based on his strong belief in social justice, George suggested that the government should take the increase in land value realized from this source in the form of a land tax (George 2006). The idea is that the benefits offered by the public should be fairly distributed among all inhabitants through mobilizing land value increment (Smolka 2013).

Although various factors contribute to land value, they are broadly categorized into 4: (1) original productivity of land, (2) private landowner's investment, (3) public investments in infrastructure and changes in land use regulations, (4) population growth and economic development (Ingram and Hong 2012; Suzuki et al. 2015). Based on this categorization, it has been suggested that landowners should obtain the increase in land value realized due to their private investment (Ingram and Hong 2012; Hong 1995), whereas it is economically appropriate for the government to capture the portion of the rise in land value attributed to public investment, urban expansion and population growth (Hong 1995; Palmer and Stephen 2015; Walters 2016; Wu et al. 2019; Noring 2019). Land value capture is an instrument by which governments apply taxes and fees to gather a certain part of the increase in land value, from the land they control, pertinent to factors other than a private investment to finance the provision of infrastructure and public services (Noring 2019; Walters 2012). There are various types of land value capture instruments. Scholars tried to group the different value capture instruments in different ways, even though it is difficult to

draw a precise line between them. For instance, Alterman (2012) divided land value capture instruments into direct, indirect, and macro. Direct value capture instruments represent betterment, capital gain tax on land and real property, annual property tax, property transfer tax, etc. Indirect value capture tools include developer obligations, exactions, impact fees or development charges, linkage fees, cost recovery, etc., and macro value capture instruments comprise nationalization of all land, long-term public leasehold, land banking, and land readjustment (Alterman 2012). Smolka (2013) also grouped land value capture instruments into three broad categories: taxes and fees, exactions and other regulatory charges, and tools used in large urban development projects. Besides, Walters (2012) classified land value capture tools as fees and tax value capture tools and nontax value capture tools. Similarly, Suzuki et al. (2015) categorized land value capture instruments into two groups: Tax-based and development-based value capture instruments. Tax-based value capture instruments include property tax, land value tax, betterment levy or special assessment, inheritance tax, capital gain tax, and transfer tax whereas development-based instruments comprise land lease, joint development, air rights sale, land readjustment, and land banking, among others. The tax-based instruments, possessing a recurrent nature, are mechanisms by which governments can collect a portion of increase in land and property value while development-based instruments, with transactional nature, are techniques through which land value capture can directly be realized using different regulatory decisions (Walters 2012; Suzuki et al. 2015). The classification given by Suzuki et al. (2015) is the most commonly used categorization.

### 3.2. Public Land Leasing and Value Capturing

Legal rights on land may be held in the form of freehold or leasehold. Land leasehold tenure refers to a contractual agreement that confers the leaseholder an exclusive right of possession of land for a limited period (Archer 1974; Farvacque and McAuslan 1992; Lai 1998). The term lease refers to the interest formed by the lease contract. In a lease agreement, the term lessor and lessee represent a landowner and a leaseholder, respectively (Farvacque and McAuslan 1992). Regarding the duration of a lease, there are land leases with a limited period, land leases with a perpetual period, and everlasting land leases with periodic revisions of the lease payment. However, the commonly applied period of land lease ranges from 35 years to 99 years as retrieved on 12 April 2020 from http://www.tink-iris.nl/downloads/public-ground-lease-and-the-effects-on-housing.pdf.

(Van Veen 2005). Land leasehold tenure may public or private. In a private land leasehold system, private parties are the owners of land rights whereas the government is the owner of those rights in a public land leasehold system (Anderson 2012). Implemented based on the philosophy of a *"bundle of rights"* (Hong 2013), the public land leasehold system is not a new concept. It existed for a long time in different parts of the world. However, the situation that the lessor is the government in the public land leasehold system makes it an innovative system. In this system, the government shares the bundle of land rights between land developers and the public (Benchetrit and Czamanski 2004). In a public land tenure system, the government has different rights on the land, as it is indicated in Figure 2. It can unpack the "bundle of rights" and transfer each right of the package to other parties, using different arrangements. The public land leasing system is one of the methods by which a government, in a public land tenure system, can transfer land use rights to other parties, keeping the ownership right in its hand (Hong 1996; Hong 1999; Blomquist 2012; Hong 2013). In addition, public land leasing is an institutional vehicle through which governments can generate revenue that enables them to provide infrastructure investments (Noring 2019).

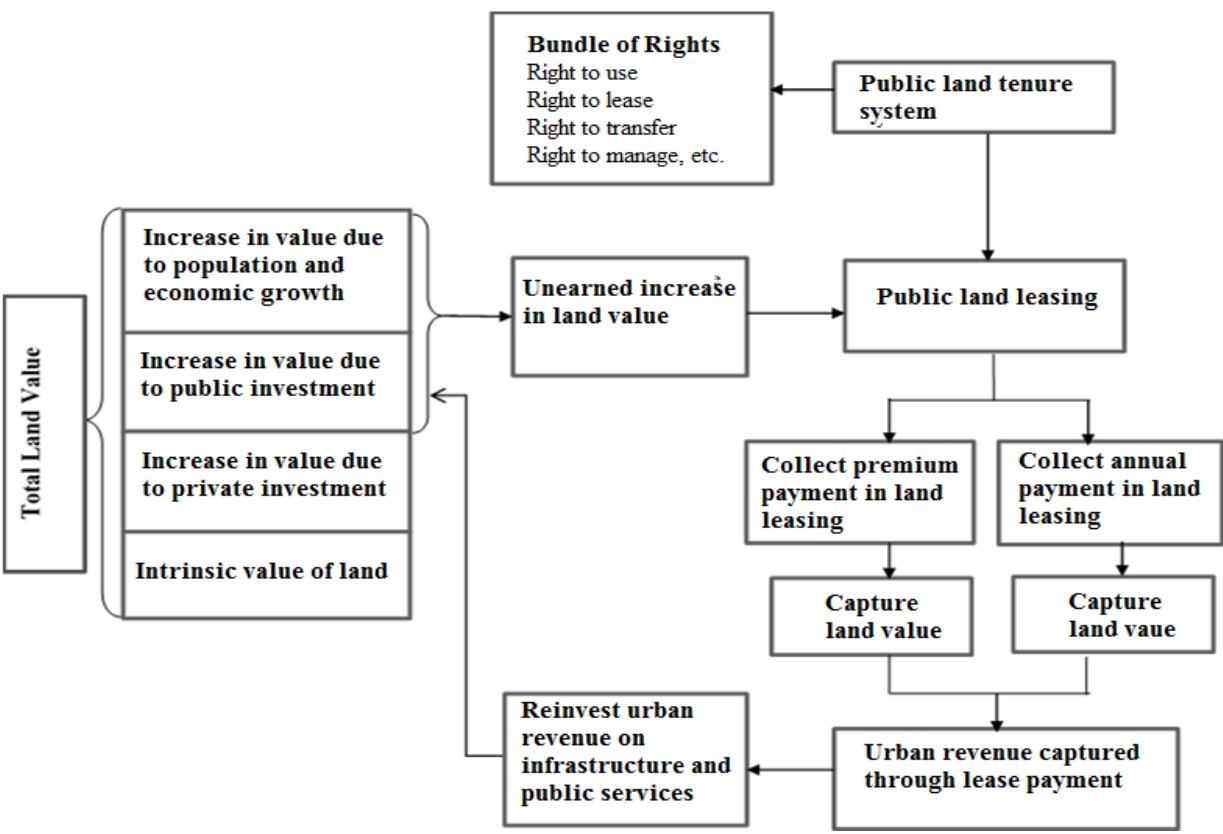

**Figure 2.** A Theoretical framework of urban land leasing as a value capture instrument in a public land ownership system. Source: Constructed by the author by blending the framework from Guyimu (2013); Maina (2014); Suzuki et al. (2015) considering countries with public land tenure systems.

In countries where the public sector owns the land, cities can lease out public land rights to capture the revenue from the increase in land value (Peterson 2007) in the form of an initial premium, annual land rent, lease modification, and lease renewals (Hong 1995; Hong 1996; Nystrom 2007). Based on the aforementioned discussions, the writers have developed theabove diagram (Figure 2) as a theoretical framework to show how governments can capture an increase in land value using public land leasing as a strategic value capture instrument in a public land tenure system. This framework has been developed by the writers of this study by blending or synthesizing the frameworks designed by Guyimu (2013), Maina (2014), and Suzuki et al. (2015), considering countries which are implementing public land tenure systems. The framework indicates that both private and public investments contribute to land value enhancement. Thus, the government could capture part or all of the enhanced land value, in the form of premium and/or annual land lease payments by implementing a public land leasing system. Using the revenue captured from an increase in land value, the government can provide infrastructure facilities and other public services, which in turn increase the land value and can be a further source of urban revenue. The figure specifically reflects the idea that in a country where there is public land ownership system, a public land leasing system could allow the government to have a virtuous cycle of land value enhancement and urban revenue generation if the government captures the land value appreciation and reinvest it to provide urban infrastructure facilities and other public services. Since Ethiopia is practicing public land tenure system, this framework is supposed to provide a theoretical and practical foundation to understand and analyze the implementation of the public land leasing system in Bahir Dar city from the perspective of land value capture.

### 3.3. Methods of Land Allocation in Urban Land Leasing System

The current literature indicates that there are different methods of land rights transfer in a public land leasing system. Auction is one of these methods and it is an advisable method of land transfer in a country, which is exposed to the perception of corruption in urban land allocation (Nystrom 2007). The other method of delivering urban land is a grant through negotiation. Countries use a grant system for providing urban land for non-profit organizations and special-purposes institutions by evaluating their project proposal (Hong 1996). However, the auction system is more market-oriented than the negotiation (Anderson 2012). In Hong Kong, the government transfers land to individual developers, on a competitive basis, through a method of public auction, and to public institutions and charities through a grant system. In both cases, land leasing is a contractual land rights allocation (Lai 1998). At the beginning of the 1990s, although Chinese municipalities used private negotiation, public auction, and private tender, private negotiation was the principal method of allocating urban land in the lease system in China (Nystrom 2007; Peterson 2007). However, later on, the government recognized that land allocation through private negotiation provided a better opportunity for corruption & malpractices (Peterson 2007; Gao 2019; Qu and Liu 2012). Consequently, the Ministry of Land Resources (MLR), in March 2002, announced Regulation No.11, granting of use rights in state-owned land by tender, auction and quotation, to be effective from 1 July 2002. The regulation stated that the transfer of all state land for business purposes should be conducted either through tender, auction, or listing (Xu et al. 2009; Zhang and Xu 2016; Dong 2017; Qu and Liu 2012). However, cities such as Beijing did not stop using negotiation in transferring land after the regulation was proclaimed. For instance, Beijing transferred more than 10,000 hectares of land through negotiation after the decree (Xu et al. 2009). Cognizant of this, the MLR enacted Regulation No. 71 on 28 March 2004 to strictly prohibit the use of negotiation as a means of land transfer. This declaration put 31 August 2004 as a deadline to stop transferring land through negotiation (Xu et al. 2009). Afterward, the auction method became the major way of leasing land for real estate development in China (Yang et al. 2015) This market-based allocation has improved the competitiveness of the land use rights market and increased the land lease revenue in China (Qu and Liu 2012).

### 3.4. Lease Benchmark Price

Studies pointed out that the Chinese government, after the introduction of the land leasing system, faced a problem in estimating the value of land due to the absence of data on the land market transaction before that period (Chan 1999; Ding 2001; Wang 2009). Consequently, the government brought a system of benchmark price (BMP) as a means to guide the transaction of land use rights (Ding and Song 2005; Ding 2001; Wang 2009; Chan 1999). As defined by the China National Standards, "benchmark land price is the within-grade average land price at a specific time point" (Wang 2009). It is not a specific land transaction price. Rather, it is an estimated average land price that simply indicates the distribution of land value for different types of land use (Wang 2009; Ai 2005). Being an average price, the benchmark land price serves as a basis on which the parties involved in land use rights transactions bargain on land price (Ding 2001; Li and Walker 1996). The assessment of the BMP should consider different factors such as land use, land grade, plot ratio, the term of lease rights, land improvement costs, land acquisition costs, municipal facilities, amenities, availability of infrastructure, costs of demolition, and tenant resettlement costs (Ai 2005; Ding 2001; Chan 1999; Wang 2009). Wang (2009) also asserted that the estimation of the BMP should consider the recent market data. The revision of benchmark land price is one of the fundamental tasks so that it could address enhancement in land value (Ai 2005). However, study findings indicated that the benchmark price was commonly lower than the actual market price of land in China due to the undeveloped land market and distortion of land market behavior by government intervention (Li and Walker 1996). Concerning the benchmark price of different land-use types, the study conducted by Li and Zhou (2017) in Shuanghe town showed that the benchmark price of commercial land

is higher than that of residential land, which is in turn higher than that of the industrial land.

### 3.5. Methods of Land Lease Payment

Hong (2013) mentioned that there are two common ways of making land lease payments. These are land premium and land rent payments. The land premium is the lump-sum amount of money paid at the beginning of the lease period, during the period of renewal of the lease, or lease modification. Land rent payment is made in the form of installments on annual basis. In the case of annual land rent payments, some countries adjust the annual lease payment periodically to capture an increase in the value of the land. For instance, Clark and Runesson (1996) mentioned that Sweden could revise lease payment with ten years difference for all land use types, as per the law revised in 1967. Van Veen (2005) stated that the land rent (lease rent) is determined based on the value of land in Amsterdam. As stated in the booklet prepared by the city of Amsterdam in 2015, the land rent is adjusted annually in line with the rate of inflation. Besides, there are 10 years and 25 years indexed ground rents that are adjusted within 10 years and 25 years respectively. The purpose of this periodic adjustment in land rent is to capture the increase in the value of land realized due to public investment in different periods (Anderson 2012).

### 3.6. Contribution of Urban Land Leasing to Revenue Generation

One of the principal objectives of urban land leasing is revenue generation, as mentioned in the current public finance literature. Related to this issue, Peterson (2007) has pointed out that Shenzhen city of China collected 80% of its local revenue from public land leasing in the 1990s; and Beijing city gathered land leasing revenue which approximates 60% of its capital expenditure from 1995 to 1996. Land leasing revenue provided financial assistance for its urbanization in China (Zhang and Xu 2016); and it helped Chinese cities nearly double their built-up areas during the period 1999 to 2009 (Ye and Wu 2014). In China, local governments started to utilize the transfer of land as a source of local revenue at the beginning of the 1990s. In 1999, the land transfer revenue only contributed 9.3% of the aggregate local budgetary revenue (Wang and Ye 2016). However, the land leasing has increased local government revenue, particularly since 2003 (Zhang and Xu 2016). Anderson and Li (2016) pinpointed that the land lease revenue contributed, on average, 55% of the cities' total revenue in China from 2003 to 2011. This share increased to 60.07% in 2011 (Wang and Ye 2016). Besides, from 1996 to 2000, the land leasing revenue surpassed its total expenditure expended on infrastructure and other public services in Hong Kong, although the land leasing revenue has varied from year to year. In Hong Kong, the land sales revenue increased from 8% in 2003 to 25% in 2013 (Cheung and Wong 2019). In addition, it has been found that land premium was one of the main government revenue sources and it contributed on average about 21% of the total government revenue, from 2015 to 2019 (The Legislative Council 2018). These indicate that urban land leasing can be a good source of urban revenue if urban governments can properly manage it.

## 4. Findings and Discussion from the Case Study

### 4.1. Forms of Urban Land Tenure and Methods of Land Leasing in Bahir Dar City

The urban land lease holding proclamation No.721/2011, which is currently under implementation, has recognized lease and permit (old possession) as the two forms of urban landholding in Ethiopia. The proclamation accepted the permit holding system, which allows the land use rights that existed during the socialist or Derg regime to continue for an unlimited period through paying nominal land rent per year. The aim of maintaining the permit system is to minimize the resistance that the government would face from the permit holders. Although the proclamation recognizes maintaining the land use rights held in the permit system, the government intends to undertake the gradual conversion of the permit system into a lease system. Related to this issue, Article 6 of the decree has mentioned how the concerned body can convert the old possessions to a lease holding

system. For instance, Article 6(1) stated that the Council of Ministers, based on a detailed study, shall decide the way of changing the old possession into the lease system. In addition, Article 6(3) of proclamation No. 721/2011 has asserted that when a property built on land held in a permit system is conveyed to another party through any form except inheritance, the party to whom the property is conveyed can possess the urban land through a lease system. Notwithstanding the provisions of Article 6, Article 5(1) of the proclamation has declared the lease system as the only way of acquiring urban land use rights in Ethiopia (FDRE 2011). In both the permit and lease system, the public is the owner of land in the country.

Although there are two principal forms of urban landholding in Bahir Dar city, it is difficult to get the exact figure regarding the total number of parcels and urban land tenure in the city due to the absence of a well-organized cadastral system. For this study, the writer has used the land parcels data with coordinate systems, which Compass AEPED Consulting firm collected and updated with the help of experts of the municipality in 2019. Accordingly, out of the 33,503 land parcels surveyed, 89% and 11% of the urban land parcels are held in permit and lease systems, respectively (MoUDH 2019). Although the exactness of the figures needs further verification, the finding, in general, indicates that most of the land parcels of the city exist under the permit system (old possession). In addition, the information obtained from key informants has strengthened the finding that the amount of land held in the permit system is higher as compared to that of the lease system. Despite its large share, the study has found that the annual rate of land rent payment per square meter on permit holding land is negligible. For instance, a permit landholder, who has the highest value (grade 1) of residential land and commercial land is supposed to pay an annual land rent with a rate of 0.17 Birr[1] per square meter and 3.00 Birr per square meter, respectively, as per ANRS Directive No.07/2006[2]. The rate is small compared with the lease benchmark price and lease tender price (See Sections 4.2 and 4.3). If a landholder has 250 square meters of highest value residential land, he/she should pay a sum of 42.5 Birr per year. This amount is meager compared with the lease payment in the land leasing system. This rate is small compared with the lease benchmark price and lease tender price, and it was estimated in 2013/14. The directive asserts that the rate should be revised within a maximum of four years. However, it has been found that the ANRS has not yet revised the rate. Consequently, the city is losing a large amount of revenue from land rent due to the small and outdated rate of land rent.

Concerning the mode of land rights transfer in the public land leasing system, Article 7(2) of proclamation No. 721/2011 has authorized regional governments and city administrations to transfer urban land in lease through public tender and allotment. However, the proclamation has declared public tender as the principal method of urban land transfer, as stated in Article 4(3) of the proclamation. The process of transferring urban land use rights through public tender should pass through different stages, as mentioned in Articles(8–11) of the proclamation (FDRE 2011).

The writer has observed that the public tender announcement has included the required information, as stipulated in the decree. Figure 3 indicates the process flow of transferring urban land to the first winner in the public tender in Bahir Dar city. The first step is the identification and preparation of urban land for tender. The last step is transferring urban land to the lessee and delivering a copy of the lease contract to the revenue office. The data obtained from annual reports of Bahir Dar city, key informants of officials, and experts have indicated that the city has been transferring its urban land to developers through public tender adhering to the process shown in the diagram (Figure 3). However, some KIIs have mentioned that there is suspicion of malpractices in finalizing the contractual agreement even though the process flow adheres to the correct mechanical procedure. Upon completing the contractual agreement, the department of house development and construction is supposed to send the lease contract to the revenue office for follow-up of the annual lease payment collection.

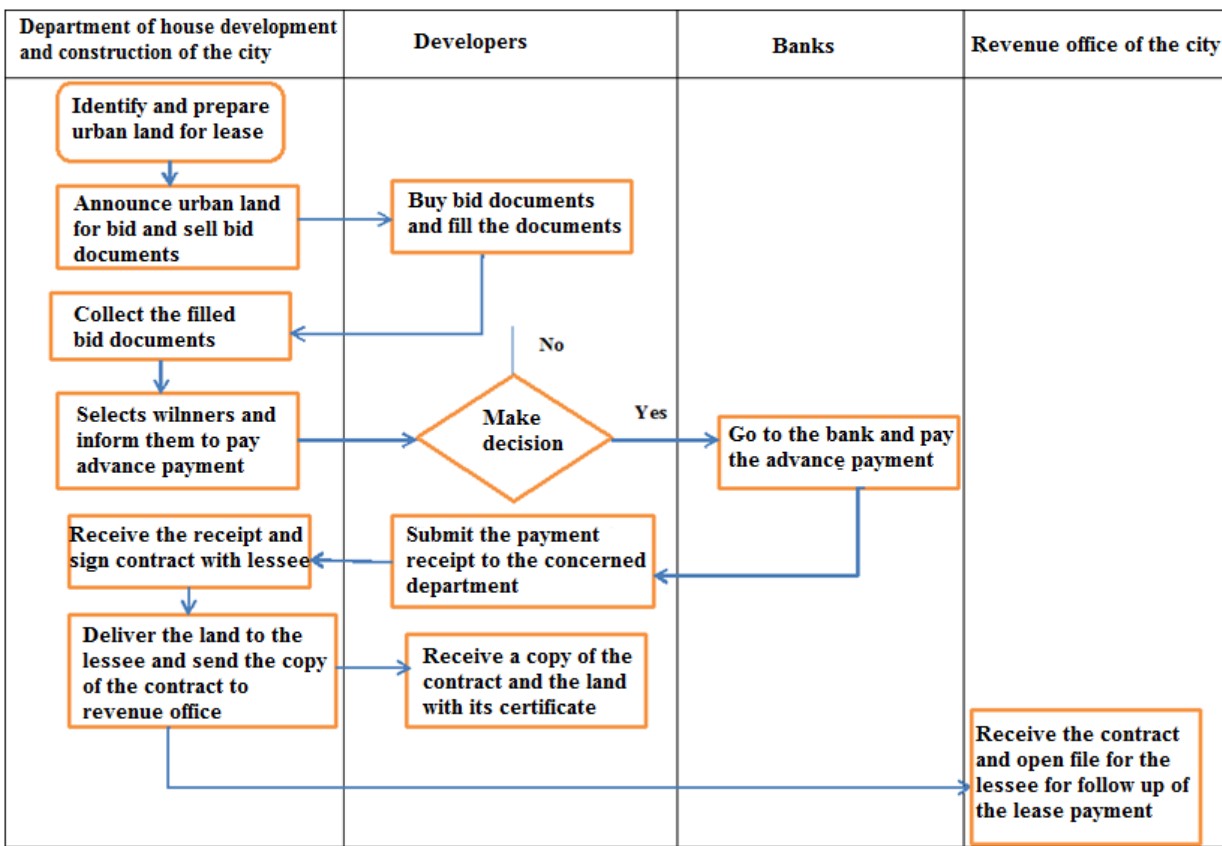

**Figure 3.** The process of urban land delivery through public tender for the first winner in Bahir Dar city. Source: Constructed by the author based on the information obtained from Bahir Dar municipality.

Although the proclamation declares public tender as the primary mode of urban land use rights transfer, the finding indicates that Bahir Dar city has been transferring most of the urban land use rights through administrative allotment. Out of the total land allocated in the city from 2013/14 to 2019/20, the amount of land delivered by public tender constitutes only 9%, which indicates that the city has allocated the majority of its urban land through administrative allotment. The land was allotted for different purposes such as to displaced landholders, manufacturing industries, housing cooperatives, government offices, charity organizations, religious institutions, and others as stipulated in the proclamation. The data indicate that the city has allotted most urban land for manufacturing industries and residential use for people organized in cooperatives. The idea is to encourage private investment in the manufacturing industry and to solve the acute shortage of residential houses in the city. Article 31(2) of the ANRS regulation No.103/2012 has stated that land allocated through administrative allotment should be transferred based on a separate price estimated for each service type or it can be transferred at a benchmark price if it is not possible to estimate a separate price (ANRS 2012). However, the study has pointed out that the city has been delivering land use rights through allotment simply at the benchmark price since it is difficult to estimate a separate lease price for each service type, which adversely affects the lease revenue-generating potential of the city. Regarding the trend of land transferred, the data indicate that the amount of land delivered by allotment has fluctuated from year to year during the period considered (see Figure 4). The key informants stated that the fluctuation occurred due to the political choice of the government in that land transfer through allotment is common during the years of government election since the government needs to maximize its political benefits.

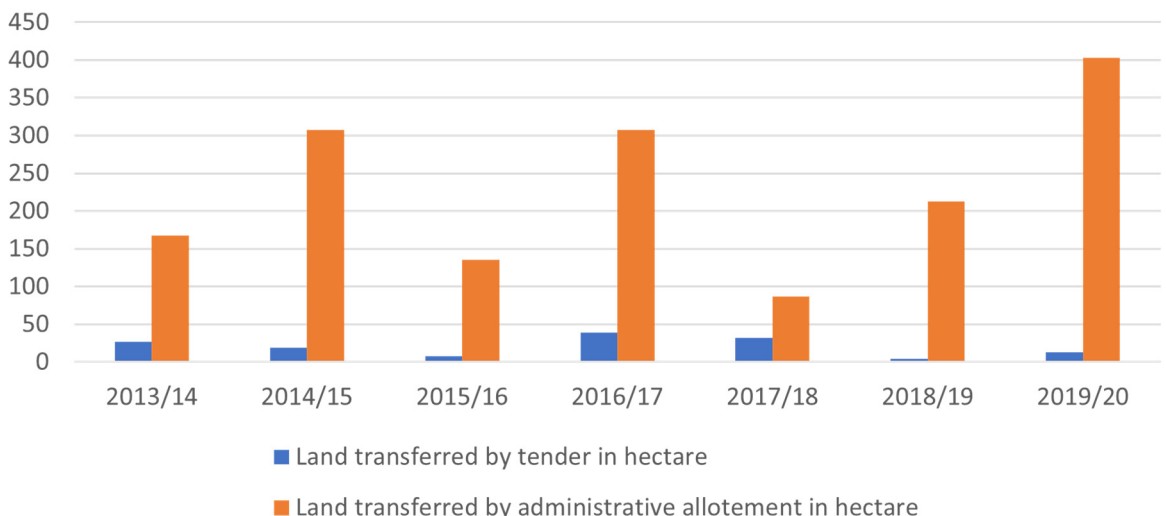

**Figure 4.** Land transferred through lease from 2013/14 to 2019/20 in Bahir Dar City in hectares. Source: Bureau of Urban and Development, Housing and Construction, 2021.

Even though the municipality has transferred land use rights as shown in Figure 4, the actual land supplied commonly lags behind its demand in the city. For instance, based on the data obtained from the annual report of the city, the municipality has planned to deliver 704 hectares of land for different purposes in 2019/20. However, it has been found that it supplied only 415.74 hectares of land this year, indicating that there is an imbalance in the demand for and supply of urban land in the city. The study has identified different factors that have resulted in slow urban land supply in Bahir Dar city. The difficulty of acquiring land, related to the land rights expropriation and compensation process, is one of the main reasons for the sluggish land supply in the city. The information obtained from key informants indicates that it is challenging to make the land free from landholders on time due to the expropriation and compensation issues. Since the amount of compensation paid to landholders is small, and it cannot enable them to sustain their livelihood, it is not easy to convince landholders to free the required land from them. The other challenge related to expropriation is that the city faces a shortage of finance to pay the compensation for landholders. In addition, the selected site for leasing may not have an approved urban land use zone. In this case, it is not easy to change the existing land use zone to the required land use due to bureaucratic hurdles. Thus, the difficulty of changing the urban land use zone deters the timely urban land supply in the city. Moreover, the study has pointed out that scarcity of finance to prepare serviced urban land hinders the timely delivery of urban land in the leasing system. Furthermore, the absence of a well-organized cadastral information system is the other issue that delays the appropriate supply of land in the city. Above all, the shortage of qualified human resources (in terms of knowledge, skill, and attitude) is the principal factor that hampers the timely delivery of urban land in the city. These institutional and structural factors have created supply-side rigidities in urban land supply in the city. This situation adversely affects the revenue-generating potential of the public urban land leasing system of the city.

*4.2. Lease Benchmark Price in Bahir Dar City*

By introducing the public land leasing system, Ethiopia started to pursue a market-oriented urban land pricing strategy to commodify urban land-use rights in 1993, with proclamation No. 80/1993. However, the emergence of this new approach became challenging at the beginning since it was a new undertaking, and there was no active land market that could be used as a reference to estimate urban land lease prices in the country, as the case occurred in China. Thus, the government brought the idea of the benchmark land price, from international experiences, which could be used for bidders as a minimum threshold

to decide on the land tender price (Goodfellow 2015, 2017), and for the government to deliver land through allotment. Related to this issue, Article 14(1) of Proclamation No. 721/2011 has declared that each plot of land should have a lease benchmark price (FDRE 2011). Besides, Article 14(3) of the proclamation has asserted that the lease benchmark price should be updated at least within two years. Addressing these provisions, Articles 29(1 and 2) of the ANRS revised urban land leasehold regulation No. 103/2012 stated the variables, which should be considered in estimating lease benchmark price. This regulation mentioned that the lease benchmark price shall be revised within each budget year or at least every two budget years (ANRS 2012). It indicates that the ANRS has understood the importance of regularly updating the benchmark price. As per the authority given by proclamation No. 721/2011 and the ANRS regulation No. 103/2012, the ANRS Bureau of Industry and Urban Development prepared directive No. 07/2006 to guide the implementation and updating of lease benchmark prices, and the rent of land held through the permit system. The Bureau organized an ad hoc committee to prepare the directive. The ad hoc committee estimated the lease benchmark price of land for all cities found in the regional state in 2013/14, and it came up with different benchmark prices for cities and towns based on the city's grade level, land use, and land grade. Table 1 indicates the lease benchmark prices in Bahir Dar city, as extracted from the directive.

**Table 1.** Lease benchmark price based on land use type and land grade in Bahir Dar City since 2013/2014 in Ethiopia Currency (Birr) per square meter (m$^2$).

| Land Grade | Benchmark Lease Price for Each Land Use Type in Ethiopian ETB Per Square Meter | | | | | | |
|---|---|---|---|---|---|---|---|
| | Residence | Commercial | Social Service | Industry | Urban Agriculture | Government Organizations[3] | NGOs |
| 1 | 350 | 900 | 390 | 60 | 50 | 100 | 80 |
| 2 | 300 | 800 | 300 | 51.42 | 45 | 90 | 70 |
| 3 | 250 | 700 | 250 | 34.28 | 35 | 80 | 60 |
| 4 | 200 | 600 | 200 | 17.14 | 20 | 70 | 50 |
| 5 | 150 | 500 | 150 | 8.57 | 10 | 60 | 40 |

Source: Extracted from Lease Benchmark Price and Land Rent Updating and Implementation Directive no. 07/2006 of ANRS (ANRS-BoIUD 2006).

Bahir Dar city comprises five different categories of land grades, as shown in Table 1. Grade one represents the highest value land, and grade five denotes the lowest value land. The classification has recognized different factors such as access to the road network, distance from the central business district (CBD), access to green facilities, etc. However, the study has understood that there is no land grade map in the city. The lease benchmark price is different for each land grade and land use type. For instance, the lease benchmark price is 350 Birr per square meter and 900 Birr per square meter for the first-grade residential land and commercial land, respectively, in the city. The data indicate that the benchmark price of commercial land is more than twice that of the residential land in each land grade type.

Regarding the method of estimation, the directive states that the benchmark prices were estimated with the objective of covering the costs of land preparation, costs of infrastructure provision, costs of compensation, and other administrative costs. It reveals that the committee applied the cost replacement method of land valuation in estimating the benchmark price. However, the directive has not stated how these costs were estimated. Anchored on the socialist doctrine of land valuation, which believes the cost of production is the bases of valuation, the estimation of the lease benchmark price has undermined the influence of the market fundamentals on land prices in the city. Consequently, the benchmark price cannot reflect the average market price of land use rights.

As per the legislation, the region was supposed to update these lease benchmark prices at least three times within the previous seven years. However, it has not yet revised

the benchmark prices. As a result, the prevailing lease benchmark prices are outdated, and they could not reflect the average market prices of land use rights in Bahir Dar city. Despite this, the city has still transferred most of its urban land use rights through administrative allotment with the outdated lease benchmark price. On the one hand, it has created a good opportunity for private land leaseholders to capture most of the increase in land value realized from public investment, population growth, and economic development. On the other hand, it has dwindled the land lease revenue-generating potential of the city and deterred its capability to finance infrastructure and other public services.

Despite it being too late; the ANRS Bureau of urban development, housing and construction formed an ad hoc committee to update the benchmark prices a year ago. The key informants, who are the members of the committee, have stated that they focus on updating the benchmark price based on the cost method of land valuation. They emphasize considering the cost of infrastructure, the cots land preparation, and the payment of compensation as the main determinant of benchmark prices although they address the land grade and land use types in their estimation. They have also pointed out they have been facing different institutional and practical challenges in updating the lease benchmark prices. The main challenges are the lack of proper coordination of the task of estimating the benchmark price by the concerned officials; the absence of professional valuation experts; the difficulty of getting the required data; the inability of identifying the appropriate land price determinants; and the absence of an independent office that manages the task. These challenges are thwarting the effort of the ad hoc committee in arriving at the appropriate benchmark prices of land in the city.

*4.3. The Trend of Urban Land Lease Tender Price in Bahir Dar*

As previously mentioned, public tender is the principal mode of urban land delivery in Ethiopia. In transferring urban land through the public tender system, the lease transaction of the price of land is determined based on the bid price and the amount of advance payment that each bidder wants to disburse (FDRE 2011). The practices in Bahir Dar city indicate that the land lease tender price submitted by bidders has a weight of 80%, and the amount of advance payment has a weight of 20% in selecting the winner of the tender as per Article 16(2) of Regulation No.103/201. The advance payment should not be less than 10% of the total lease payment as stipulated in the proclamation. The winner of the tender is decided based on the sum of these two weights. A bidder that scores the highest sum of these weights can be the winner of the land lease tender. According to the regulation, if two bidders score the same result and one of them is female, priority will be given to the female and the lease rights will be awarded to the female competitor, by which the regulation tries to be gender-sensitive. Otherwise, the final decision shall be decided based on a lottery method (ANRS 2012). It can be understood from this finding indicates that a bidder who offers the highest land lease tender price per square meter may not necessarily be the winner of the bid since the possibility to win the competition depends on both the lease tender price per square meter and the amount of advance payment that the lessee will pay. Table 2 shows the city's minimum, average, and maximum land lease tender price of residential and commercial land registered from 2013/14 to 2019/20. The lease tender price stated in the table has excluded some outliers to avoid the distortions they may create in comparing the average lease tender prices among themselves and with the lease benchmark prices.

**Table 2.** Range of actual tender land lease price per square meter in Bahir Dar city. 2013/14–2019/20 in Ethiopian Currency (Birr).

| Year | Price Residential Land in Ethiopian Birr/Square Meter | | | Price of Commercial Land in Birr/Square Meter | | |
|---|---|---|---|---|---|---|
| | Minimum | Mean | Maximum | Minimum | Mean | Maximum |
| 2013/14 | 841 | 3126 | 7016 | 403 | 3757 | 16,006 |
| 2014/15 | 3800 | 6251 | 11,731 | 1619 | 5751 | 15,101 * |
| 2015/16 | 1466 | 13,323 | 22,110 | 910 | 4291 | 15,501 |
| 2016/17 | 9101 | 15,580 | 25,000 | 1151 | 7222 | 28,550 |
| 2017/18 | 7555 | 19,759 | 45,000 | 4100 | 10,266 | 25,600 |
| 2018/19 | 14,200 | 26,602 | 39,005 | 3927 | 16,966 | 30,005 |
| 2019/20 | 30,000 | 39,254 | 56,200 | 6002 | 23,851 | 28,650 |

Source: Extracted from public lease tender documents of Bahir Dar Municipality. * We found one share company that won the lease with a lease price of 71,500 Birr/m$^2$ in 2014/15. However, it has been excluded in the calculation of the average and maximum lease price since it is an outlier.

This study has pointed out that the average land lease tender price of residential land has steadily increased, from 2013/14 to 2019/20, although the benchmark prices that belong to this period remain constant. The same is also true for the average lease tender price of commercial land, as indicated in Table 2 and Figure 3. The key informants have identified the following four principal reasons for the land lease tender price hike up in Bahir Dar city. Firstly, the government monopoly of land supply in the primary land use rights market creates a shortage of land in the city, which commonly creates intense competition among bidders to get land out of the limited number of parcels announced in the bid. Secondly, the population of Bahir Dar city has been rapidly increasing although the supply of land is fixed. This situation has resulted in an imbalance in the demand for and supply of urban land, which has in turn pushed up the market price of land in the city. Thirdly, the urban land delivery system is inefficient due to a lack of budget to service the urban land, lack of finance for compensation, shortage of qualified experts, and malpractices in the land management system as mentioned in the previous section. These factors deter the timely supply of urban land in the city and further increase the price of urban land. Fourthly, most people in the city perceive holding urban land as the principal means of accumulating wealth. Thus, they may not hesitate to offer a higher land bid price to win the public tender and take the urban land through the lease system since they speculate that they will sell it at a higher price in the future. The fifth reason is the rent-seeking behavior of brokers in the process of land transactions. After a lessee gets land through the leasing system, he/she may informally sell it to other persons. The transaction process often takes place through a broker. As the broker's commission increases with the price of land, the broker commonly sells the leasehold rights at a higher land price. Thus, lessees are not afraid to offer a higher land lease tender price to win the competition with the expectation of a higher land lease rights transfer price with the help of a broker.

Figure 5 indicates that the average lease tender price of residential land and commercial land has steadily increased during the study period. However, the average lease tender price of commercial land is lower than the average lease tender price of residential land each year mainly starting from 2016/17. It seems that it is contrary to the benchmark price and the theory of urban land price. The key informants have provided the following justifications for this trend. Firstly, the size of commercial lands announced through public tender was larger than the size of residential land in the city. Thus, offering a higher bid price per square meter increases the amount of the advance payment. As a result, lessees may not be encouraged to give a higher lease price per square meter for commercial land. Secondly, a rise in the demand for housing in the city increases the demand for residential land; and it, in turn, intensifies the competition for residential land. Thirdly, most of the urban land leased through public tender locates in the expansion areas where commercial activities do not flourish. As a result, lessees, who need land for commercial use, may not be motivated to offer higher bid prices per square meter in such locality. The study has

uncovered that these factors have resulted in, on average, a higher residential land lease price per square meter than that of commercial land in the city.

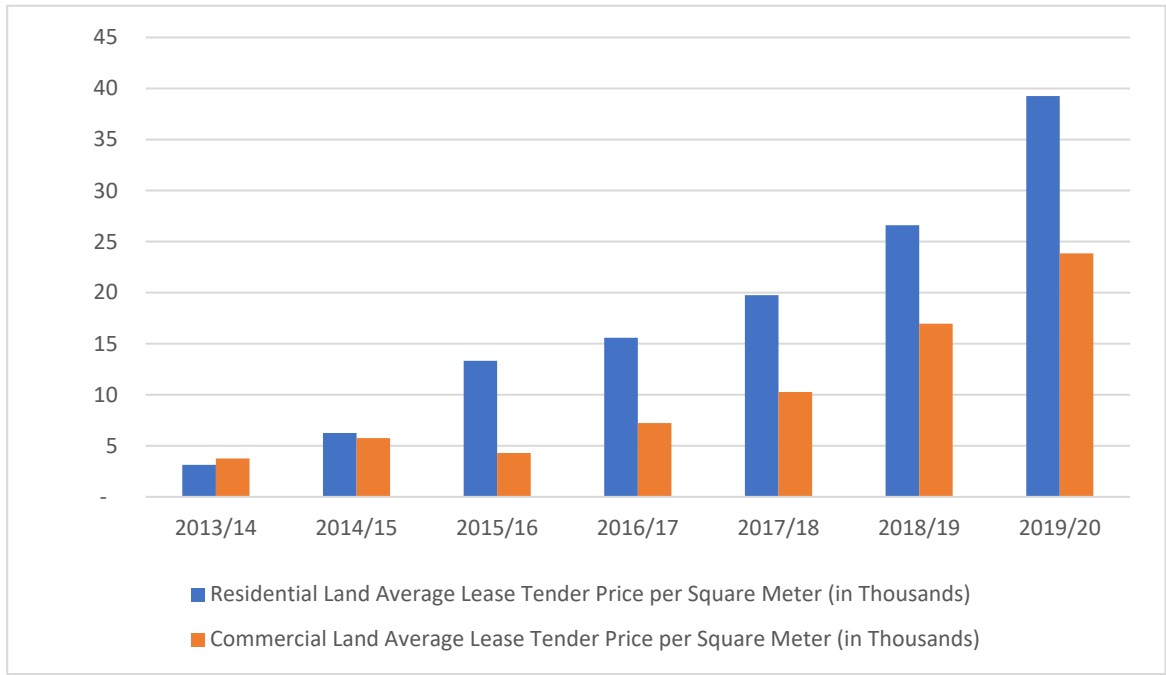

**Figure 5.** The trend of average lease tender price in Bahir Dar city from 2013/14–2019/20 in Birr. Source: Bahir Dar City Municipality, 2021.

Regarding the relationship between the lease tender price and the lease benchmark price, the finding shows that the average tender prices significantly deviate from the lease benchmark prices of land in Bahir Dar city, as shown in Tables 1 and 2 above. For instance, the average tender price of residential land was 3126 Birr/m$^2$, and that of commercial land was 3757 Birr/m$^2$, although the benchmark lease price of the first-grade residential land and commercial land was 350 Birr/m$^2$ and 900 Birr/m$^2$ in 2013/14, respectively. The comparison of these figures shows that the lease tender price of the residential land was 9 times higher than the maximum lease benchmark price of land even in 2013/14. Besides, the average of the 7years' average residential land tender prices, 17,699 Birr/m$^2$, is 50 times the lease benchmark price of the highest value (first-grade) residential land. Moreover, the data obtained from brokers indicate that the price of land in the expansion area of Bahir Dar city, where most of the leasehold land locates, ranges from 7000 Birr/m$^2$ to 12,000 Birr/m$^2$ in the informal secondary land use rights market, which is widespread in the city. This meant that the average market price of land is about 9500 Birr per square meter. This price is considerably higher than the lease benchmark price of the highest value land. These findings indicate that the benchmark prices do not reflect the average market price of land in the study area, and there is a wide gap between the lease tender prices and the lease benchmark prices of land in Bahir Dar city. As a result, the city has been losing a large amount of revenue from leasing land through administrative allotment since the city has transferred most of its urban land through allotment at the outdated lease benchmark price. The existence of this large difference indicates that the factors that could result in this deviation need further empirical investigation so that the empirical evidence will support the future determination of benchmark prices in Bahir Dar city.

### 4.4. Methods of Land Lease Payment and Value Capture

Article 32 of Regulation No.103/2012 states that a lessee has to pay an advance payment, not less than 10% of the total lease payment, and an annual lease payment. The

annual lease payment can be calculated by dividing the total lease payment (remaining after deducting the advance payment) by the lease payment period, which is a maximum of 50 years for residential land, 40 years for other land uses, and seven years for urban agriculture. In addition, the lessee has to pay interest, calculated based on the Commercial Bank of Ethiopia loan interest rate, on the annual lease payment. A lessee has to pay the advance payment before signing the lease contract. However, he/he should start to pay annual lease payments and interest payments after completing the grace period offered for the lessee (ANRS 2012). This meant that Bahir Dar city collects lease revenue in the form of advance payments, annual rent payments, and interests on annual rent payments. The regulation also asserts that there is a discount system for lessees that can pay all lease payments upfront. According to Article 32(2) of Regulation No. 103/2012, for instance, a lessee who can disburse all of the lease payments upfront in 1 year can obtain a discount of 5% of the total lease payment. The region has designed this payment modality to motivate an upfront payment that can allow municipalities to generate immediate revenue from land leasing. Although the ANRS has designed this motivational package, the data obtained from the city has shown that most of the lessees have not used this opportunity. The reason mentioned by the key informants is that the total lease payment is high, and it is difficult for lessees, mainly for those who got the land through tender, to pay the entire lease payment as a one-time upfront payment.

Regarding value capture, the study found that the city is not able to capture the land value increase over time since the lease proclamation does not permit revision of the lease price once the lease contract is signed. In Ethiopia, the land use rights market is categorized as primary and secondary. According to Article 24(1) of proclamation No. 721/2011, lessees are authorized to transfer their leasehold rights to the extent of the lease payment paid. The secondary market, in which lessees transact their lease rights, may be formal or informal. In transferring the land lease right in the secondary market, however, the law allows a lessee to take the sum of the lease payment made together with its interest, the value of improvements developed on the land, and five percent of the total transfer value of the leasehold rights. In this process, the government shall capture the remaining or extra amount of the transfer value of the lease rights. The idea is that it is unjustified for the lessee to capture all of this extra value. The proclamation has authorized the municipality to regulate the secondary land use rights market and capture the unjustified gains realized during the transfer of land use rights in the secondary market. The municipality can easily capture the unjustified gain if the lessee transfers his/her leasehold rights in the formal secondary land use rights market since such markets are regulated by municipalities (see Figure 6). However, the study has pointed out that the informal secondary market is very active in the city since the higher profit usually motivates the leaseholders to participate in this market.

According to the data obtained from key informant interviews (KIIs), most informal land use rights transactions are undertaken in such a way that the lease rights holder transfers the right to the other developer as if he/she borrows a certain amount of money from the other developer taking the leasehold rights as collateral. The leaseholder commonly undertakes such transactions through brokers by making informal agreements with the buyers of the land use rights and thereby capturing all of the total transfer value of leasehold rights. The formal government system does not enable the municipality to capture most of the unjustified gains earned by lessees since the administrative system is not capable enough to avoid the abusive actions of the hidden nexus among lessees, brokers, and buyers of land lease rights. Consequently, the municipality has lost a large amount of revenue related to the land use rights transfer. The data obtained from the interview have revealed that institutional weakness and loopholes in the legal framework are the main factors that are responsible for this phenomenon.

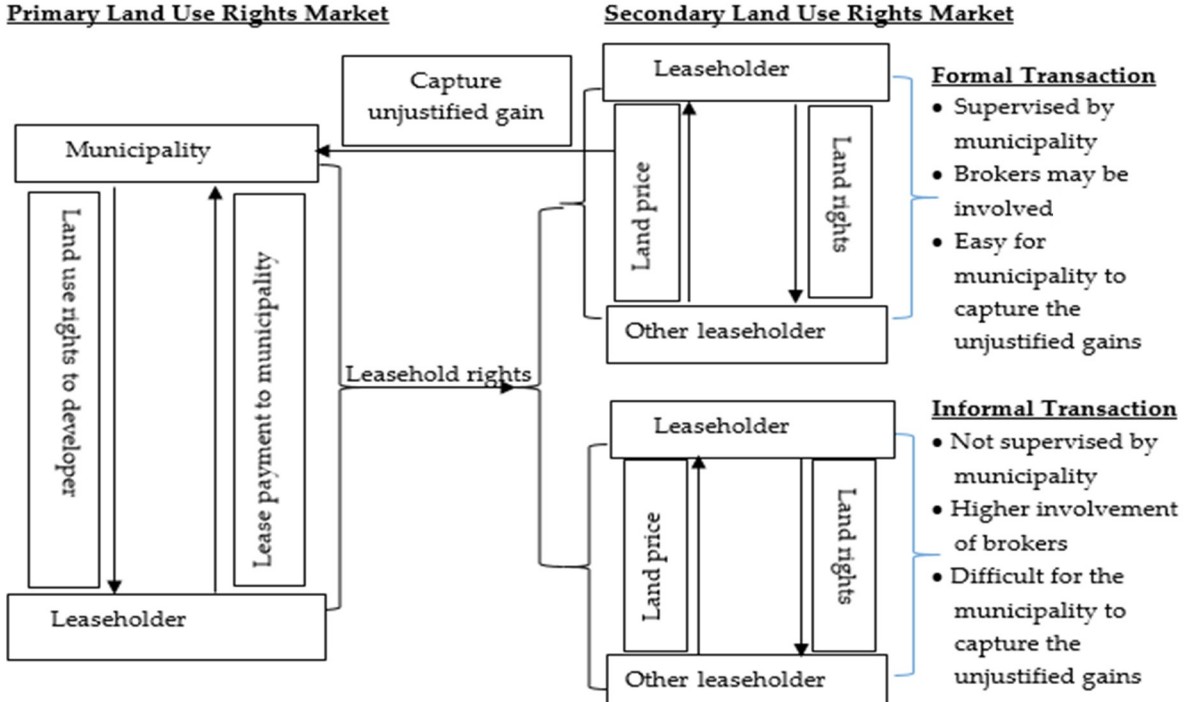

**Figure 6.** Urban land leasehold rights transaction in Ethiopia. Source: Constructed by the author by modifying the figure mentioned in (Wu and Zheng 2011) considering the transaction of urban land leasehold rights in Ethiopia.

*4.5. Enforcement of Land Lease Payment Collection in Bahir Dar City*

The data obtained from the Bahir Dar city revenue office indicate that the actual revenue collected from land leasing is, on average, more than 90% of the planned revenue in the previous 7 years. In some years, the office has gathered an income beyond its plan. For instance, in 2019/20, the office has planned to collect a revenue amounting to 250 million Birr from land lease payments and it has collected about 292 million Birr. In contrast to this result, the researcher has observed a letter written, for the third time within 1 year in 2019/20, from the revenue office to the department of housing construction and development of the city requesting it to take appropriate action on default lessees. The letter has included the list of 326 delinquent lessees with a total accumulated land lease payment arrears amounting to 108.8 million Birr. The number of default lessees might be greater than this since there is no well-organized cadastral information in the office. It has been found from the letter that there are lessees who have not paid the lease payment for ten consecutive years. So, the question that must be raised is how could the city collect the revenue beyond its annual plan given this situation? The data obtained from the key informants indicate that this is because the city has allocated more land for cooperatives that were waiting to received residential land for several years, and it has collected advance payment from them during 2019/20. It was this injection in revenue, which boosted the city's annual performance of lease revenue collection without a predetermined plan that year. In addition, it was understood that the revenue office commonly understates its collection plan due to the absence of modern cadastral information on lessees in the city.

The study has also pointed out that the enforcement of lease payment collection is not robust for the following reasons as per data obtained from the KIIs. First, there is no well-organized file management system on urban land possession in the city. As a result, the revenue office could not follow up on the lease payment collection. Second, there are only two experts in the revenue office who are currently responsible for handling the lease payment collection although the task needs to have a separate department with the appropriate number of experts. Third, the revenue office and the department of housing

construction and development are accountable to two different offices. This structural problem has resulted in the weak horizontal linkage between these two offices. As a result, it has become difficult for the revenue office to make the housing construction and development office accountable for enforcing the lease payment collection as per the contract. This study has found this structural fragmentation as the main bottleneck that has impeded the enforcement of lease payment collection in Bahir Dar city. Third, the shortage of employees and leaders with the required knowledge, skill, and attitude has been mentioned as the other reason that deters the enforcement of lease payment collection.

As a panacea to this problem, most of the key informants have suggested organizing a separate department, which is responsible for municipal revenue collection under the revenue office. The department will comprise different teams, and one of them will be held accountable for land lease revenue collection. This team will have regular communication with the housing construction and development office. However, some of the key informants have suggested organizing a municipal revenue collection department parallel to the department of housing construction and development under the municipality so that both will be accountable to the same office. It will also enable the municipal revenue collection department to provide single window customer services for lessees. The other idea suggested was thinking out of the box and outsourcing the lease payment collection function to unemployed graduates by organizing them under a cooperative. It can have a twin objective of improving the efficiency of the lease payment collection on the one hand and creating job opportunities on the other hand. However, this requires further study and benchmarking of the experiences of other countries regarding its management.

*4.6. Contribution of Land Lease Revenue to Total Municipal Revenue of the City*

In a study carried out in some cities in the Amhara and Tigray regions of Ethiopia, (Peterson 2007) found that land lease revenue contributed about 45% of Bahir Dar city's total revenue in the year from 2003 to 2004. In addition, (MoUDH 2016) has pointed out that the city's land lease revenue contributed more than 58% of its total municipal revenue from 2010/11 to 2014/15. This study has also found that, out of the total municipal revenue of the city, the land lease revenue has contributed, on average, approximately 56% from 2013/14 to 2019/20. It varies from 48% to 67% in these years. The land lease revenue has increased during this period although it has fluctuated from year to year. Figure 7 indicates that the city's total revenue has fluctuated following a similar pattern to the change in land lease revenue. This pattern has reflected the condition that the land lease revenue of the city has had a conspicuous influence and contribution to its total municipal revenue during the period under consideration. The positive correlation between lease revenue and total municipal revenue reveals that the sale of land-use rights has benefited the city and it could benefit more than this if it managed the system properly.

This study has pinpointed that land leasing revenue has accounted for a larger share of the total municipal revenue of Bahir Dar city. As per the information obtained from the KIIs, this is because of the following reasons. First, the amount of revenue, which the city has been collecting from lease advance payment, has increased due to the rapid rise in the lease tender price, per square meter of the urban land. Second, the city government has provided a large amount of land for cooperatives and manufacturing industries, through administrative allotment, in the years under consideration. From this, the municipality has collected the lease payment as an upfront premium, which has augmented the contribution of land lease revenue to the total municipal revenue of the city.

Although land leasing is the cardinal source of municipal revenue in Bahir Dar city, this study has pointed out that the city has not tapped its land lease revenue-generating potential up to its maximum capacity due to the different factors discussed in the previous sections.

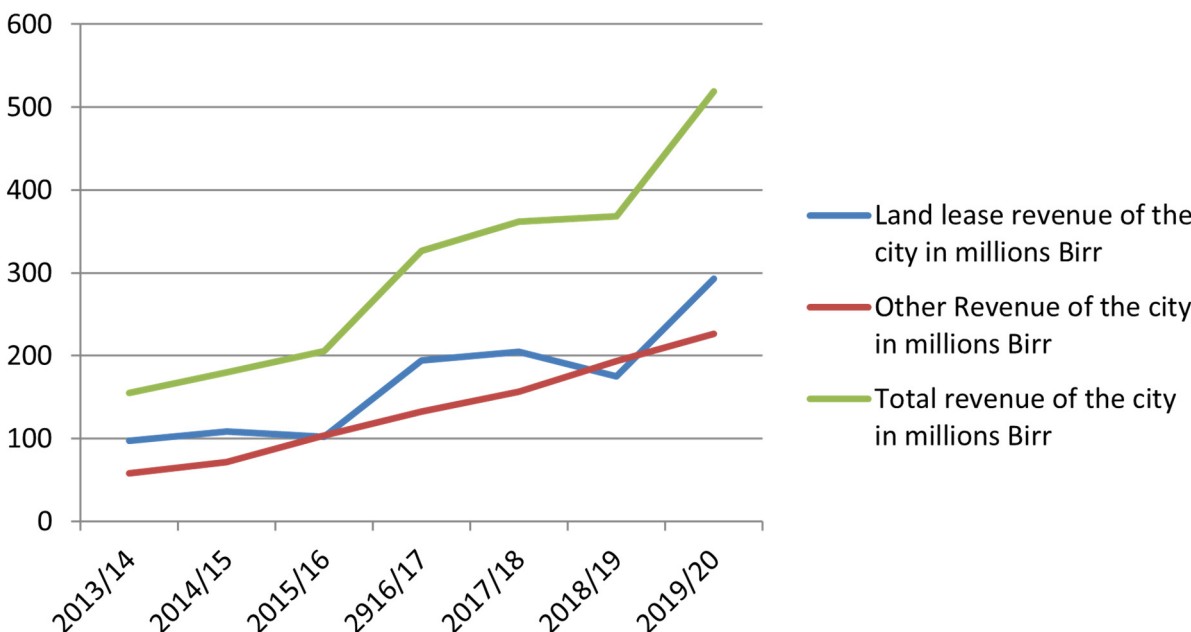

**Figure 7.** The relationship between total revenue and lease revenue of Bahir Dar city. Source: Extracted and complied from annual reports of the Revenue Office, and Finance and Economic Cooperation Office of Bahir Dar City.

## 5. Conclusions and Policy Implications

This study tries to explore and understand the contribution of the public urban land leasing system as a strategic valuate capture instrument and the existing problems that constrained the system in urban revenue generation in Bahir Dar City. The following conclusions are drawn from this study. First, although the urban lease holding proclamation No. 721/2011 recognizes lease holding and permit holding as the two forms of urban landholding in Ethiopian cities (FDRE 2011), the study has identified that most of the urban land is possessed through the permit system in Bahir Dar city. In a permit system, land users maintain the land use rights for an unlimited period in exchange for a nominal amount of land rent per year. The land rent is, however, calculated based on an outdated and negligible amount of rate per square meter. Therefore, this phenomenon compels the city to lose a large amount of land-related revenue.

Second, the study has identified a supply-side rigidity in delivering the urban land in the city mainly due to the difficulty of acquiring land, lack of finance to provide serviced land, bureaucratic hurdle in changing land use zone, shortage of qualified human resources, and the absence of well-organized cadastral information system. This means the city could not supply the required amount of land, and it dwindles the revenue-generating potential of the public urban land leasing system in the city.

Third, the study has also found that the lease benchmark prices, which were estimated seven years ago based on the socialist doctrine of replacement cost valuation method, do not reflect the average transaction price of land. In addition, the city has not yet updated the benchmark prices even though it was supposed to do it at least three times during this period, according to the legislation. Given this status quo, the city has been allocating most of the urban land through administrative allotment for different purposes at the outdated benchmark prices. This situation has diminished the revenue that the city could collect from public urban land leasing. It has been adversely affecting the city's capability to finance the provision of infrastructures and other public services.

Fourth, in transferring land lease rights, the lease proclamation has authorized the public to capture the extra land value created due to changes in land use, public provision of infrastructure, and population growth in the city since it is unjustified for lessees. However, the study has found that the city could not practically stop lessees from appropriating

this unjustified gain realized from land value appreciation. The reason is that lessees, in collaboration with brokers and buyers of the lease rights, are 'gaming the system' to garner all transfer value of the lease rights. The administrative system is not capable enough to avoid such abusive actions of the hidden nexus among lessees, brokers, and buyers of land lease rights. This leads to a considerable loss in urban revenue and in social injustice unless the government acts appropriately.

Fifth, it has been pointed out that the city cannot change the land lease price with a change in the value of the land after it signs the lease contract with the lessee. This meant the city could not capture the future increase in land value through the lease system.

Finally, the study has found that the land lease revenue has on average increased from 2013/14 to 2019/20 even though it has fluctuated from year to year. Land leasing revenue has had a conspicuous contribution to the total municipal revenue of the city. Despite this noticeable role of land lease revenue in enhancing municipal revenue, this study has identified accumulated land lease payment arrears for several years, in Bahir Dar city, because of the ineffective lease collection system and weak enforcement of the lease contracts. To sum up, the study has shown that Bahir Dar city is not utilizing the public urban land leasing system up to its optimal potential as a strategic value capture instrument to enhance its local revenue.

In view of these findings and conclusions, this study has forwarded the following suggestions.

- To capture the value increment on the permit holding land, the government should implement property tax or re-estimate the rate of land rent considering the factors that mainly affect the value of land, and regularly make it up-to-date so that it can allow the city to gather the proper amount of land revenue;
- In leasing land using administrative allotment, the city should use an estimated current land transaction price or a regularly updated benchmark prices, which can reflect the mean land transaction prices so that the city can earn an appropriate amount of revenue from land leasing;
- Since the lease benchmark prices considerably deviate from the mean lease tender transaction prices in the city, the government should empirically identify the main determinants of the lease tender price and update the lease benchmark price accordingly within the timeframe stipulated in the lease proclamation;
- To capture the future increase in land value, the government either should attach the annual lease payment with the rate of inflation, or it should regularly appraise the land value, and update the land lease payment accordingly at least at three or five years intervals;
- To minimize the appropriation of the unjustified gain from the land value appreciation by private land use rights holders through informal land transactions, the government should develop a cadastral information system and control the land transaction activities with this system. In addition, it is important to develop a system that can improve people's attitudes towards formal land transactions;
- To realize the timely supply of land in the city, the study argues that the city administration has to provide the appropriate amount of compensation for evicted landholders and improve the capacity of the municipality in terms of human resources (knowledge, skill, and attitude), financial, and other required resources;
- The government has to install a modern cadastral land information system in the city to have reliable land information since it can assist the municipality to manage the public land leasing system and enforce lease payment collection;
- Since different regional states may have different regulations and directives under the federal proclamation, and urban land lease price is mainly affected by location factors, this study may not reflect the overall situation that exists in other cities throughout the country. Thus, it is important to carry out a similar study in other cities as well so that national policymakers can have a comprehensive understanding of the issue.

**Author Contributions:** S.H.Y. has carried out the conceptual design, methodology, data collection, and data analysis; H.L. has supervised the study, and B.Y.A. has co-supervised the study. All authors have read and agreed to the published version of the manuscript.

**Funding:** This research was sponsored by the German Academic Exchange Service (DAAD) through its In-Region/In-Country scholarship programme in collaboration with the GIZ assisted programme "Strengthening Capacities for Land Governance in Africa (SLGA)". The grant's personal reference number was 91642092.

**Conflicts of Interest:** The authors declare no conflict of interest.

## Notes

[1]  Birr represents an Ethiopian currency. One USD = 19.074 Birr in 2013/2014, 1USD = 20.0956 Birr in 2014/15, 1USD = 31.34 Birr in 2019/20 based on the date obtained from the Annual Report of National Bank of Ethiopia, and 1USD = 40.1537 Birr on 4 March 2021 based on the data obtained from https://nbebank.com/commercial-banks-exchange-rate/ (accessed on 4 March 2021).

[2]  The year 2006 is in the Ethiopian calendar, and it is 2013/14 in the Gregorian calendar.

[3]  Offices for budgetary organizations, government schools and health institutions, and religious institutions are free from lease payment except paying the amount to replace the compensation payment paid to acquire the land from previous landholders according to this directive.

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
