# Peer review of "Understanding Urban Land Leasing System as a Strategic Value Capture Instrument to Enhance Urban Revenue in Ethiopia: A Case Study of Bahir Dar City"

_economies, doi:10.3390/economies10060146_

Round 1
Reviewer 1 Report
This paper aims to explore whether the Bahir Dar city uses the urban land lease system as a strategic value capture tool to increase its local revenue. This study adopts qualitative research method and in-depth analysis. It gathers the data needed to achieve the objectives of this study through a desk review of documents and key informant interviews with experts and brokers. My comments are as follows.
1. What are the differences between this study and other similar publications related to land lease system and land finance? What is the contribution of this study?
2. The general background and policy of land lease system in Bahir Dar city are missing.
3. In Line 322, listing of quotations, tender, and auction to allocate land use rights of the public-owned land through a leasehold system probably stipulated in 2004, and not in 2002. Please have a check.
4. Some references are too out-of-date, such as the literature in 1996, 1999, etc. The references in rent years are missing. If there is no reference in rent years cited in this paper, how can author grasp the gap of the literature, and prove the innovation of this paper? Some literature in rent years should be added in literature review section, and the literature related to this study also should be added in this section.
5. In Section 3.6, the data of land leasing revenue in 1990s is relatively early. Is there any data of land leasing revenue in 2000s?
6. For the title of Section 4.1, it refers to ‘land transfer’, but in other sections or other paragraphs, there refer to ‘land lease’. Please unify which word to use.
7. This research focuses on Bahir Dar city. Can this study give a reference to other cities or countries?
8. This paper lacks the analysis and explanations of some issues. For example, in Section 4.6, land lease revenue accounts large proportion to total municipal revenue of the city, but there is no analysis of why this phenomenon happens.
9. The study mentioned interview is one of the research methods. In this paper, there show less information on interviews. Can author summarize the content of the interviews, the specific interviewees, and the conducting time of the interviews?
Author Response
Thank you for your review and constructive comments. We read your comments carefully and made corrections as described below point by point. .
Point 1: What are the differences between this study and other similar publications related to land lease system and land finance? What is the contribution of this study?
Response 1: We acknowledged this point. We have addressed the issue by revising the “introduction” section of the paper in such a way that the difference between this study and other similar studies, the research gap, and the contribution of the study have been clarified. The part which shows these revisions reads as follows:
“The reviews of literature show that most of the studies (Yusuf et al., 2009; Alemu, 2010; Gebremariam and Mailimo, 2016; Belete, 2017) have paid attention to assessing the impact of the urban land lease policy on real estate investment, private sector business expansion and property rights in Addis Ababa and the others (Adamu, 2014; Tigabu, 2014; Ambaye, 2016) are national-level studies. However, Goodfellow (2017) has tried to analyze public land leasing from the perspective of value capture although he has focused on Addis Ababa and it might not reflect the situation in Bahir Dar city due to differences in location and socioeconomic conditions between the two cities. In addition, Teferi (2009) has not exhaustively addressed the factors that may affect the urban land leasing system in generating local revenue. To sum up, the previous studies have paid little attention to urban land leasing from the context of value capture in Ethiopia. As a result, there are no adequate documented scientific studies carried out on the extent to which a city like Bahir Dar city has utilized urban land lease as a strategic value capture instrument to enhance its local revenue although it is an essential public policy issue. These circumstances, therefore, can justify undertaking an in-depth investigation and detailed analysis to understand the implementation of the urban land leasing system from the perspective of value capture and revenue enhancement in Bahir Dar city.”
“Thus, the purpose of this study is to explore and understand whether Bahir Dar city is leveraging the urban land lease system as a strategic value capture instrument to enhance its local revenue or not. The main contribution of this study is to provide decision-makers and practitioners with an in-depth understanding and some scientific insights for improving the existing urban land leasing system as a strategic value capture instrument. Based on this objective, this research has addressed the following questions: (1) How can the urban land leasing system enable the government to capture the upsurge in land value realized from public investment, economic growth, and population growth in Bahir Dar city? (2) Is the city leveraging the urban land leasing system to enhance its local revenue? (3) What are the main factors that deter the city from enhancing its revenue related to public land leases? (4) What are the actions that should be taken to enable the system to improve the city’s local revenue?”
Point 2: The general background and policy of land lease system in Bahir Dar city are missing.
Response 2: Bahir Dar city does not have a separate urban land lease policy. The country has an urban land lease policy, which should be followed by Bahir Dar and other cities in the country. This policy has been described in the introduction section, which read as follows:
“After it came into power in 1991, the Transitional Government of Ethiopia (TGE) found that the urban revenue collected from land rent, house taxes, and other sources was not adequate to finance the provision of infrastructure and other public services. Besides, it realized that the existing law could not enable municipalities to monetize the increase in land value created due to public investment and economic growth. Consequently, the TGE introduced the urban land leasing policy in 1993, as per proclamation no. 80/1993, which aimed to enable urban centers to improve their local revenue and monetize the land value appreciation unjustifiably collected by private landholders, among its other objectives (FDRE, 1993). With this reform, the TGE separated land ownership rights from land use rights in the urban areas. Afterwards, the government has undertaken a sequence of amendments to the urban land lease proclamation. The first proclamation allowed public tendering as the only mode of urban land leasing (FDRE, 1993). Since the proclamation did not apply to the urban land held during the Derg regime(FDRE, 1993), two different forms of urban land tenure (i.e., permit and lease) co-existed in the country, which led to “tensions between de jure and de facto rights of urban land”(Adamu, 2014). In addition, studies found that the tender method resulted in a rigid urban land supply (Yusuf et al., 2009). Thus, the government enacted proclamation no. 272/2002 in 2002 to fill the existed gaps, and it replaced proclamation no.80/1993 (Alemu, 2010; Adamu, 2014). The new proclamation stipulated how to convert the urban land held through a permit system into a lease system (FDRE, 2002). Besides, the decree introduced negotiation as an additional method of land leasing. However, the negotiation method opened opportunities for corrupted practices (Yusuf et al., 2009). Studies (Yusuf et al., 2009; Adamu, 2014) show municipalities experienced ineffective, corrupted, and less transparent land delivery practices. Due to this and other compelling reasons, the government replaced proclamation no.272/2002 with proclamation no.721/2011, considering good governance as the fundamental prerequisite to realize an efficient, transparent, and well-functioning land market (Adamu, 2014).”
Point 3: In Line 322, listing of quotations, tender, and auction to allocate land use rights of the public-owned land through a leasehold system probably stipulated in 2004, and not in 2002. Please have a check.
Response 3: We have acknowledged this issue and we have addressed it by including additional information, which read as:
“Ministry of Land Resources (MLR), in March 2002, announced Regulation No.11, granting of use rights in state-owned land by tender, auction and quotation, to be effective from July 1, 2002. The regulation stated that the transfer of all state land for business purposes should be done either through tender, auction, or listing (Xu et al., 2009; Qu and Liu, 2012; Zhang and Xu, 2016; Dong, 2017). However, cities such as Beijing did not stop using negotiation in transferring land after the regulation was proclaimed. For instance, Beijing transferred more than 10,000 hectares of land through negotiation after the decree (Xu et al., 2009). Cognizant of this, the MLR enacted Regulation No. 71 on 28 March 2004 to strictly prohibit the use of negotiation as a means of land transfer. This declaration put 31 August 2004 as a deadline to stop transferring land through negotiation (Xu et al., 2009). Afterwards, the auction method became the major way of leasing land for real estate development in China (Yang et al., 2015)”
Point 4: Some references are too out-of-date, such as the literature in 1996, 1999, etc. The references in rent years are missing. If there is no reference in rent years cited in this paper, how can author grasp the gap of the literature, and prove the innovation of this paper? Some literature in rent years should be added in literature review section, and the literature related to this study also should be added in this section.
Response 4: We have acknowledged your concern. We addressed it by adding recent literature related to this study, which read as follows: (Perkins, 2009; Teferi, 2009; Xu et al., 2009; Yusuf et al., 2009; Blanco Blanco et al., 2016; Wang and Ye, 2016; Belete, 2017; Dong, 2017; Gebremichael, 2017; Weldesilassie and Gebrehiwot, 2017; Yin, 2018; Rashid et al., 2019; Gebrihet and Pillay, 2020)
Point 5: In Section 3.6, the data of land leasing revenue in 1990s is relatively early. Is there any data of land leasing revenue in 2000s?
Response 5: We have acknowledged this issue. We have addressed it by including land lease revenue data in the 2000’s, which reads as follows:
“In China, local governments started to utilize the transfer of land as a source of revenue at the beginning of the 1990s. In 1999, the land transfer revenue contributed only 9.3% out of the aggregate local budgetary revenue (Wang and Ye, 2016). However, the land leasing increased local government revenue, mainly since 2003 (Zhang and Xu, 2016). Anderson and Li (2016) pinpointed that the land lease revenue contributed, on average, 55% of the cities' total revenue in China from 200 to 2011. This share increased to 60.07% in 2011(Wang and Ye, 2016).”
“In Hong Kong, the land sales revenue increased from 8% in 2003 to 25% in 2013 (Cheung and Wong, 2019). In addition, it has been found that land premium was one of the main government revenue sources and it contributed on average about 21% of the total government revenue, from 2015 to 2019 (Council, 2018).”
Point 6: For the title of Section 4.1, it refers to ‘land transfer’, but in other sections or other paragraphs, there refer to ‘land lease’. Please unify which word to use.
Response 6: We have recognized this comment and we have unified the two phrases according to your comment.
Point 7: This research focuses on Bahir Dar city. Can this study give a reference to other cities or countries?
Response 7: We understood your concern. Although the research focuses on Bahir Dar city, its findings can be used as a reference for other cities in the country since the policy and the legal frameworks, on which they base their tasks, are similar. It can also be used as a reference to other countries with similar legal frameworks and policy set up.
Point 8: This paper lacks the analysis and explanations of some issues. For example, in Section 4.6, land lease revenue accounts large proportion to total municipal revenue of the city, but there is no analysis of why this phenomenon happens.
Response 8: We have accepted your comment. We have addressed it by including the reasons why land lease revenue contributed to the total municipal revenue of the city more than other revenue sources, which reads as:
“This study has pinpointed that land leasing revenue has accounted for a larger share of the total municipal revenue of Bahir Dar city. As per the information obtained from the KIIs, this is because of the following reasons. First, the amount of revenue, which the city has been collecting from lease advance payment, has increased due to the rapid rise in the lease tender price, per square meter of the urban land. Second, the city government has provided a large amount of land for cooperatives and manufacturing industries, through administrative allotment, in the years under consideration. From this, the municipality has collected the lease payment as an upfront premium, which has augmented the contribution of land lease revenue to the total municipal revenue of the city.”
Point 9: The study mentioned interview is one of the research methods. In this paper, there show less information on interviews. Can author summarize the content of the interviews, the specific interviewees, and the conducting time of the interviews?
Response 9: We acknowledge your comment. We have improved the methods section by including additional clarifying information, which read as:
“The primary data were collected from first-hand sources, i.e., from key informants, using the semi-structured interview guides. Two forms of interview guides were prepared: one for experts and officials of government offices, and the other for brokers in the real estate market. The interview guides addressed issues such as the process of land leasing, estimation of benchmark prices, collection of lease payments, land use rights transactions, etc. A total of 18 interviewees were purposively selected, i.e., 15 from Amhara National Regional State (ANRS) bureau of urban development and construction, ANRS Bureau of revenue, the urban revenue reform project Bahir Dar branch office, and different relevant departments of Bahir Dar city municipality, and 3 from real estate brokers in the city. The interviewees were selected considering their experience. The face-to-face key informants’ interviews were administered by the authors and takes two to three hours each.”

Reviewer 2 Report
Let me thank the author(s) for letting me review their work. The paper is well written and suitable for publication in the journal Economics. The paper is consistent and fits in with the overall journal scope. The lit. rev. and references are sufficient. Some additional referencing are needed in the Conclusions and Policy Implications section.
The research idea is good and its significance is somewhat novel; as it stands the novelty is medium. The paper length is fine. The paper is written clearer—I would recommend another read through for the English. The theoretical framework is sound; the results are also clear and extensive.
The paper is relevant and interesting and shows promise. Sufficient background into the topic has been presented and I would recommend the paper for publication.
ENGLISH: The paper should be carefully revised for English typographical mistakes. Some parts do not read clearly. In general, I would like to thank the authors for submitting a nicely constructed paper. A few minor revisions will make the paper publishable.
Author Response
First of all, the authors would like to thank you for reviewing and recommending the paper to be published. Then, we addressed some of your minor comments as follows.
Point 1: The paper should be carefully revised for English typographical mistakes. Some parts do not read clearly.
Response 1: We acknowledged your comment. So, we have gone through the document and corrected the typographical mistakes.
Point 2: Some additional referencing are needed in the Conclusions and Policy Implications section.
Response 2: We have acknowledged your comment and we have added the required references.

Round 2
Reviewer 1 Report
The authors have fully responded to my comments.